# Potential Modulation of Inflammation by Probiotic and Omega-3 Supplementation in Elderly with Chronic Low-Grade Inflammation—A Randomized, Placebo-Controlled Trial

**DOI:** 10.3390/nu14193998

**Published:** 2022-09-27

**Authors:** Lina Tingö, Ashley N. Hutchinson, Cecilia Bergh, Lena Stiefvatter, Anna Schweinlin, Morten G. Jensen, Kirsten Krüger, Stephan C. Bischoff, Robert J. Brummer

**Affiliations:** 1Nutrition-Gut-Brain Interactions Research Centre, School of Medical Sciences, Örebro University, 70362 Örebro, Sweden; 2Food and Health Programme, Örebro University, 70362 Örebro, Sweden; 3Division of Inflammation and Infection, Department of Biomedical and Clinical Sciences, Linköping University, 58183 Linköping, Sweden; 4Clinical Epidemiology and Biostatistics, School of Medical Sciences, Örebro University, 70362 Örebro, Sweden; 5Department of Nutritional Medicine and Prevention, University of Hohenheim, 70599 Stuttgart, Germany; 6GSK Consumer Healthcare ApS, 2665 Vallensbæk Strand, Denmark; 7Human Nutrition & Health, Department of Agrotechnology and Food Sciences, Wageningen University & Research, 9101 Wageningen, The Netherlands

**Keywords:** elderly, chronic low-grade inflammation, probiotics, omega-3, *n*-3 PUFA, hs-CRP, RCT, short-chain fatty acids

## Abstract

Probiotic and omega-3 supplements have been shown to reduce inflammation, and dual supplementation may have synergistic health effects. We investigated if the novel combination of a multi-strain probiotic (containing *B. lactis* Bi-07, *L. paracasei* Lpc-37, *L. acidophilus* NCFM, and *B. lactis* Bl-04) alongside omega-3 supplements reduces low-grade inflammation as measured by high-sensitivity C-reactive protein (hs-CRP) in elderly participants in a proof-of-concept, randomized, placebo-controlled, parallel study (NCT04126330). In this case, 76 community-dwelling elderly participants (median: 71.0 years; IQR: 68.0–73.8) underwent an intervention with the dual supplement (*n* = 37) or placebo (*n* = 39) for eight weeks. In addition to hs-CRP, cytokine levels and intestinal permeability were also assessed at baseline and after the eight-week intervention. No significant difference was seen for hs-CRP between the dual supplement group and placebo. However, interestingly, supplementation did result in significant increases in the level of the anti-inflammatory marker IL-10. In addition, dual supplementation increased levels of valeric acid, further suggesting the potential of the supplements in reducing inflammation and conferring health benefits. Together, the results suggest that probiotic and omega-3 dual supplementation exerts modest effects on inflammation and may have potential use as a non-pharmacological treatment for low-grade inflammation in the elderly.

## 1. Background

### 1.1. Low-Grade Inflammation in the Elderly and Inflammaging

As life expectancy in developed countries increases, it is estimated that the elderly population will consume 75% of health care resources by 2030 [1]. Therefore, identifying factors to improve health, quality of life, and independence for the elderly is essential to reduce healthcare costs and societal burden, as well as to promote individual well-being. As people age, they become more susceptible to disease and disability. However, this increased susceptibility could be reduced or prevented by adequately addressing specific risk factors (WHO). Aging is associated with complex changes and dysfunction of the immune system, including the components that regulate inflammation. Acute inflammation is a vital component of the immune response and involves the activation and recruitment of chemical messengers (cytokines, etc.), antibodies, and immune cells in response to injury or infection [2]. Although acute inflammation is necessary to repair damaged tissue and prevent infection, inflammatory processes can have negative health consequences if sustained and prolonged. Chronic states of systemic low-grade inflammation have been associated with a variety of conditions and health outcomes, including aging. The chronic low-grade inflammation associated with aging is characterized by an increase in the concentration of pro-inflammatory markers in the blood (including IL-6, IL-8, C-reactive protein (CRP), tumor necrosis factor (TNF), and others)—a phenomenon that has been termed “inflammaging” [3,4]. Such increased levels of aging-associated pro-inflammatory markers can be detected in the majority of elderly, even in the absence of a clinical diagnosis [5,6]. Inflammaging is, however, also highly implicated in many chronic conditions in older adults including increased susceptibility to infections, altered body composition, increased risk for cardiovascular disease, and impairments in physiological and physical function [7]. The mechanisms underlying inflammaging are still poorly understood, although several studies have indicated genetic susceptibility, accumulation of senescent cells, imbalance between the production and disposal of cellular debris, and immune cell defects as having potential roles [7].

### 1.2. The Use of Nutritional Supplements to Reduce Low-Grade Inflammation

In recent years, there has been a great interest in utilizing nutritional supplements to reduce low-grade inflammation. Two supplements that have been investigated are probiotics and omega-3 supplements.

#### 1.2.1. Probiotics

Probiotics are defined as “live organisms that, when administered in adequate amounts, confer a health benefit on the host” [8]. Probiotics work to stabilize gut commensal microbiota and provide resistance to colonization by pathogenic bacteria, thereby improving states of dysbiosis [9,10]. In the elderly, probiotic supplementation has been shown to improve intestinal barrier function in a number of gastrointestinal conditions including constipation, ulcerative colitis, and irritable bowel syndrome (IBS) [10]. In addition to increasing the resistance towards pathogens and pathobionts, probiotics have been also shown to have immunomodulatory effects and anti-inflammatory action [9,10,11]. For example, Moro-Gacria et al. found that a six-month intervention with *Lactobacillus (L) delbrueckii* spp. *Bulgaricus* 8481 resulted in decreases in the inflammatory cytokine IL-8 [12]. In addition, a crossover study could show that Soluble Corn Fiber in combination with *L. rhamnosus* GG-PB12 (pilus-deficient derivative) resulted in decreased CRP levels after a three-week intervention [13]. Furthermore, the efficacy of probiotics to reduce inflammation may have potential use in treating Alzheimer’s disease (AD) and cognitive impairment [14]. Although the majority of these studies included small numbers of participants, the findings suggest that probiotic supplementation may be efficacious in reducing inflammation and improving immune function in elderly [15].

#### 1.2.2. Omega-3

Omega-3 (*n*-3) is one of the two major classes of polyunsaturated fatty acids (PUFAs) together with omega-6 (*n*-6) fatty acids. *n*-3 PUFAs are rich in certain foods such as flaxseed and fatty fish, as well as dietary supplements such as fish oil. Several different *n*-3 PUFAs exist, but most of the scientific research focuses on three: alpha-linolenic acid (ALA), eicosapentaenoic acid (EPA), and docosahexaenoic acid (DHA) (NIH 2018). From a mechanistic point of view, EPA and DHA, together with arachidonic acid (AA; 20:4 *n*-6), are substrates in the synthesis of lipid mediators (e.g., eicosanoids) that are involved in inflammatory processes. In these processes, the immune mediators produced from DHA and EPA shift the immune balance toward resolution of inflammation [16].

In addition to their immune system effects, omega-3 may have effects on the intestinal barrier and physical function [17,18]. In experimental mucositis mice models, omega-3 supplementation has been shown to restore intestinal barrier function and decrease intestinal permeability and bacterial translocation [17]. Studies in humans evaluating the effects of omega-3 on intestinal barrier function are limited. However, some studies have been performed to assess the effects of omega-3 on microbiota composition, and these studies suggest that omega-3 may act as a prebiotic, shifting microbiota composition towards beneficial SCFA-producing bacteria such as *Bifidobacterium* (*B*) and restoring eubiosis [18]. Improvement of dysbiosis may thereby lead to improved barrier function and decreased inflammation. In sarcopenic patients, omega-3 supplementation has resulted in potential benefits in increasing muscle mass, muscle strength and physical performance through reduction of anabolic resistance as a proposed mechanism [19].

Furthermore, EPA and DHA have been associated with improved muscle composition or muscle strength in older adults [20,21,22] and lower intake of omega-3 is cross-sectionally associated with poor physical function [22,23]. Omega-3 is also thought to have an analgesic effect in arthritis as a likely consequence of its high content of DHA and EPA; hence, supplementation with EPA- and DHA-rich oil could exert an anti-inflammatory effect, making it a possible treatment for, e.g., arthritis pain [24]. Taken together, omega-3 has significant potential as a supplement for alleviating inflammation and the associated health complications in the elderly.

#### 1.2.3. Dual Supplementation

As outlined above, both probiotics and omega-3 fatty acids have been shown to have similar effects on immune regulation and intestinal barrier function. A systematic review and meta-analysis showed that both supplements significantly reduce both IL-6 and CRP levels when administered alone [25]. However, very few studies have thus far examined the effects of dual supplementation [26]. For example, administering omega-3 (sourced from krill oil), probiotics (*L. reuteri*), and vitamin D in mouse models of intestinal inflammation improved epithelial function by reducing inflammation, improving repair responses, and reducing pathogenic effects of harmful bacteria [27]. In addition, a clinical study evaluating the effects of a probiotics (VSL#3) and omega-3 fatty acids on insulin sensitivity, blood lipids, and inflammation demonstrated that VSL#3 and omega-3 administered alone reduced hs-CRP; this effect was even more pronounced when the supplements were administered together [28].

### 1.3. Significance and Aim

Although the results are promising, very few studies with only small sample sizes have investigated the effects of nutritional supplements, such as probiotics and omega-3 supplementation, on low-grade inflammation in the elderly. Dual supplementation may provide additional health benefits. For example, as mentioned above, omega-3 may act as probiotic, a potential mechanism underlying the synergistic effects of dual supplementation [18]. We hypothesized that the novel combination of a multi-strain probiotic enriched with omega-3 will decrease low-grade systemic inflammation in elderly participants, and, in turn, have beneficial effects on intestinal permeability and physical function.

In the present study, an eight-week intervention with a multi-strain probiotic (*B. lactis* Bi-07, *L paracasei* Lpc-37, *L. acidophilus* NCFM, *B. lactis* Bl-04) alongside omega-3 supplements was administered with the aim to lower low-grade inflammation in elderly participants. To our knowledge, no clinical trial to date has evaluated the combined effect of probiotics and omega-3 on low-grade inflammation as a primary outcome in this population. Hence, this two-armed, proof-of-concept study was designed to provide initial insights into the potential efficacy of dual supplementation to reduce inflammation in an elderly population. In addition, we also explored the effects of the supplements on gastrointestinal permeability, short-chain fatty acids in stool, and physical function as secondary and exploratory outcomes.

## 2. Materials and Methods

### 2.1. Ethics Approval

The study was performed according to the Helsinki declaration and its revisions and was approved by the Central Ethical Review Board of Uppsala, Sweden (registration number 2018/509). The trial is registered at ClinicalTrials.gov under ID: NCT04126330 and was performed at Örebro University in Örebro, Sweden.

### 2.2. Design

An eight-week randomized, double-blinded, placebo-controlled parallel-group study was performed in order to explore the effects of the combination a multi-strain probiotic and omega-3 (with vitamin D) on low-grade inflammation in elderly subjects. The primary outcome was the effect of the dual supplementation on hs-CRP levels. Secondary outcomes included the effect of the study products on anti-inflammatory and pro-inflammatory cytokines, barrier function as assessed by the multi-sugar urine recovery test, the Western Ontario and McMaster Osteoarthritis Index (WOMAC), sit-to- stand test, fasting blood glucose and insulin levels, omega-3/omega-6 ratio in blood, and intestinal fatty-acid binding protein (I-FABP) levels. Exploratory outcomes included a Food Frequency Questionnaire (FFQ), vitamin D blood levels, and fecal short-chain fatty acid (SCFA) analysis.

Prior to any study-related procedures, all potential participants received information about the study and signed an informed consent form. At the screening visit (visit 1), height and weight measurements were obtained for BMI calculations, and blood samples (approximately 4 mL) were obtained for hs-CRP measurement. Elderly with hs-CRP levels of 1.5–6 mg/L and BMI values of 18.5–27 kg/m^2^ were included in the study and randomized to either the supplement intervention or placebo. Participants were stratified based on sex so that equal numbers of men and women were allocated to each arm. Prior to the baseline visit, the participants came to pick up instructions and material for the sugar permeability test and fecal sample collection kit. At baseline (visit 2), fasting blood samples were obtained for the analyses of hs-CRP and other blood markers. Participants filled out the WOMAC questionnaire and the FFQ. In addition, the participants also completed the sit-to- stand test. The participants left their urine and fecal samples for analysis. The participants also received a sufficient number of capsules for the eight-week intervention; both the participants and the study teams were blinded to the treatment group. The participants were asked to begin taking the capsules the day after visit 2 and received a chart to log their compliance. At week four (visit 3), fasting blood samples were obtained for the analyses of hs-CRP and other blood markers. Participants received material for collection of urine and fecal samples to bring with them to the week eight visit (visit 4). At the end of week seven, as close as possible to the study visit in week eight, the participants collected urine and fecal samples at home. Finally, at week eight (visit 4), fasting blood samples were obtained for the analysis of hs-CRP. On the same day, the participants handed in urine and fecal samples, filled out the WOMAC, completed the sit-stand test, and filled in a questionnaire if they had undergone any significant changes in diet throughout the study. They also handed in their compliance charts as well as the capsule bottles. At each visit, adverse events and changes in medication were recorded. Refer to Figure 1 for an overview of the study visits and the measures performed.

### 2.3. Subjects

Subjects were recruited by advertisement in Örebro in the autumn of 2019 and the winter of 2020. Inclusion criteria included 65–80 years-old at the start of the study, a signed informed consent prior to any study related procedures, a blood hs-CRP value of 1.5–6 mg/L at the screening visit, and a normal weight at the screening visit as defined by a BMI range of 18.5–27 kg/m^2^. Furthermore, the participants needed to be willing to abstain from regular consumption of probiotic supplements and medication known to alter gastrointestinal functions for at least four weeks prior to inclusion in the study. Finally, the participants needed to be willing to fast for at least five hours, receive a standardized breakfast, and drink at least 1.5 L of water in one day for the multi-sugar urine recovery test. Exclusion criteria were chosen in order to recruit a healthy elderly study population with low-grade inflammation. A list of the exclusion criteria can be found in Appendix A. Participants were instructed not to change their diet, medication, and exercise habits throughout the study.

### 2.4. Study Intervention

The study product consisted of either two dietary supplements or corresponding placebo products. The first supplement contained both omega-3 and vitamin D with total dosage (2 capsules) of 1100 mg fish oils and 200 IU of Vitamin D. The 1100 mg fish oil consisted of 640 mg *n*-3 PUFA, -of which 300 mg eicosapentaenoic acid (EPA) and 220 mg docosahexaenoic acid (DHA) (VNP Active *n*-3 PUFA manufactured by Pfizer CH). The second supplement was a four-strain probiotic combination based on a commercially available formulation. Participants consumed a total dosage of 2 capsules per day, containing 10 billion colony forming units (CFUs) in total. Each capsule contained equal amounts of the four bacterial strains per capsule (1.25 × 10^9^ CFU each), which included two strains of *Bifidobacterium* (*B. lactis* Bi-07, *B. lactis* Bl-04), and two strains of *Lactobacillus* (*L. acidophilus* NCFM and *L. paracasei* Lpc-37) (Bifiform Daily Plus, supplied by Pfizer CH, manufactured by Danisco, DuPont). The control group received maltodextrin capsules for the probiotic placebo and capsules with sunflower oil for the omega-3 placebo.

Participants were required to consume four capsules per day for a total of eight weeks. The products were to be taken in the morning in conjunction with breakfast at approximately the same time each day, with or without food, but had to be consistent for each individual throughout the study. Participants maintained their typical diet throughout the intervention period.

### 2.5. Randomization and Blinding

Eligible participants were randomly assigned to each study arm (50% treatment and 50% placebo). The randomization was carried out using a computerized randomization list and block randomization with a random block size of six and four was used. Sex stratification ensured that equal numbers of men and women were allocated to each arm. The masking in the study was “quadruple”, i.e., participants, as well as research staff, primary investigator and outcome assessors, were blind to the intervention allocation until the study was finalized. The code key was held by a third party who was not involved in the study.

### 2.6. Study Outcomes

#### 2.6.1. Blood Sample Collection and Analysis

All blood samples were collected the morning after an overnight fast at the same time of day and were either analyzed the same day (hs-CRP, glucose, and insulin) or stored at −80 °C until analysis (inflammatory markers, I-FABP, *n*-6/*n*-3 PUFA, vitamin D). Refer to Appendix A for more detailed information on the protocols utilized for blood sample collection and analysis.

#### 2.6.2. In Vivo Intestinal Permeability Test

Intestinal permeability was assessed using a multi-sugar urinary recovery test of orally administered water-soluble sugars of different sizes: lactulose, sucralose, sucrose, rhamnose and erythritol. This is a sensitive, non-invasive technique able to detect subtle changes in small and large intestinal permeability [29,30]. The sugar molecules larger in size (lactulose and sucralose) only pass the intestinal barrier paracellularly, and no active uptake occurs. The smaller-sized molecules (rhamnose and erythritol) pass the barrier by transcellular passage and provide a control for gastric emptying and dilution, intestinal transit time, absorption area, systemic distribution and renal function. The sugar recovery ratios from an early urine fraction (urine collection of 0–5 h after intake of sugars) and a later fraction (urine collection from 6–24 h after sugar intake) are then utilized to assess the intestinal permeability of the respective intestinal segments where absorption of the sugars occurs [31,32,33,34].

Urine for the multi-sugar permeability test was collected at home at baseline and after the intervention. After one night of fasting, the participants consumed 150 mL of tap water containing 1 g sucrose (Nordic sugar, Malmö, Sweden), 1 g lactulose (Solactis, Jouy-en-Josas, France), 1 g sucralose (Bulk Powders, Sweden), 1 g erythritol (Ingredi, Stockholm, Sweden) and 0.5 g rhamnose (BioGaia, Stockholm, Sweden) at 8.30 in the morning. The night fast (commenced at 10 pm after eating a dinner that was identical between both test occasions) was interrupted 6 am by a standardized breakfast (i.e., a 55 g chocolate bar of the brand OneWhey, containing 20 g protein, 9.4 g fat and 15.5 g carbohydrates (for additional details of nutritional content refer to Appendix A), before which the participants had emptied their morning void. The standardized breakfast was followed by another 2 h 30 min fasting before the sugar solution was ingested, as described above. After drinking the sugars, the participants were asked to collect all urine in two urine jars (3000 mL, Sarstedt, Helsingborg, Sweden), stored in cooling bags equipped with ice packs for 24 h. The first jar contained all urine from 0–5 h and the second jar all urine from 5–24 h. During collection of fraction 1 (0–5 h), the participants were asked to maintain their fasting state and to consume at least 1.5 L of water. During the 5–24 h urine collection, the participants were allowed to eat, but had to refrain from alcohol, nicotine, caffeine and food items containing any of the four sugars: lactulose, sucralose, rhamnose and erythritol. After finishing the 24 h collection, both urine fractions were returned to the staff. Upon receiving the jars, urine was aliquoted into 4 mL V-monovette urinary tubes (Starstedt, Helsingborg, Sweden) and stored at −80 °C. For analysis, the urine samples were thawed, and 1 mL aliquots were centrifuged at 21,000× *g* for 25 min at 4 °C. The supernatant was collected and used for analysis by High Performance Liquid Chromatography Mass Spectrometry (HP-LCMS) to detect urinary excretion ratios of the sugars.

### 2.7. Sample Preparation and Determination of Sugar Concentration

5 μL of urine sample was transferred to Liquid Chromatography vials and 5 μL of 13C labelled internal standard (Sigma Aldrich, St. Louis, MO, USA) was added. The urine aliquots were diluted to a volume of 500 ul with a composition of 80:10 acetonitrile: water. Before analysis, all samples were vortexed for 10 s and then centrifuged for 15 min at 8000× *g*. Analysis was conducted on an Acquity UPLC coupled to a Quattro Premier XE UPLC–MS/MS system (Waters Corporation, Milford, MA, USA) with an atmospheric electrospray interface operating in negative ion mode. For further details refer to Roca Rubio (2021) [35].

The estimated sugar recovery was then used as an indicator of permeability in the following gastrointestinal segments: gastroduodenal: sucrose (0–5 h fraction); small intestine: lactulose/rhamnose (L/R) (5–19 h); large intestine: sucralose/erythritol (S/E) (5–24 h fraction); whole gut permeability: sucralose (0–24 h fraction).

#### 2.7.1. Short-Chain Fatty-Acids in Stool Samples

Fecal samples were collected by the participants in their home, following a standardized protocol and using sterile material. Refer to Appendix A for more detailed information regarding the protocols utilized for sample preparation and analysis.

#### 2.7.2. Sit-to-Stand Test (SST)

The sit-to-stand test (SST) was conducted as previously described at baseline (w0) and after the intervention (w8) [36]. Participants were instructed to sit in a straight-backed armless chair with a hard, flat seat; the floor-to-seat height was approximately 48 cm. Participants were asked to sit in the church and come forward until their feet were flat on the floor with arms folded across the chest. Next, participants were asked to stand up and sit down once without using their arms. For those unable to complete the initial procedure, the test was considered not passed. If the participants passed the initial test, they were then asked to stand up and sit down as quickly as possible. The time it took to stand up and sit down five times was recorded (in seconds) and was considered the participant’s score [36].

#### 2.7.3. Western Ontario and McMaster Osteoathritis Index (WOMAC)

Activity levels were assessed using the Western Ontario and McMaster Osteoarthritis Index (WOMAC) questionnaire at baseline (visit 2) and study end (visit 4). WOMAC can be used to detect physical disability and has been originally developed for patients with osteoarthritis of the hip and/or knee and serve as a measurement of the participant’s mobility [37]. WOMAC consists of 24 items divided into three subscales: (1) pain (during walking, using stairs, in bed, sitting or lying, and standing), (2) stiffness (after first waking, later in the day), (3) physical function (stair use, rising from sitting, standing, bending, walking, getting in/out of a car, shopping, putting on/taking off socks, rising from bed, lying in bed, getting in/out of bath, sitting, getting on/off toilet, heavy household duties, light household duties). Each item is self-reported on a scale of 0–4 (no symptoms-severe symptoms), giving a range of subtotals for pain (0–20), stiffness (0–8) and physical function (0–68).

#### 2.7.4. Food Frequency Questionnaire (FFQ)

For the elderly, the dietary pattern was assessed through a Food Frequency Questionnaire (FFQ)—Meal-Q [38]. This instrument is validated for Sweden and provides information regarding subjects eating patterns during the past 12 months. The questionnaire includes 174 foods and drinks and estimates the dietary pattern for one year. It took approximately 20 min to complete and was filled out at baseline. Participants were asked about any major dietary modifications made during the study once the study was completed.

### 2.8. Statistical Analysis

#### 2.8.1. Power Calculation

Based on our main aim of a 15% reduction of hs-CRP after the intervention, we calculated that 88 participants were required for this intervention. This power calculation was performed on basis of previous data generated in our research group. A standard deviation (SD) of 0.25 for log-level hs-CRP can be expected and together with a maximum drop-out rate of 10%, a significance level of 5% and 90% power, we estimated that 44 participants in each arm were needed; a total of 88 subjects randomized to either treatment of placebo.

#### 2.8.2. Data Analysis

In line with the primary outcome to demonstrate a 15% outcome in hs-CRP levels, percent change values (referred to as % change pre-post in all result tables) were determined for all variables. The percent change corresponds to the change between the study end visit (week eight (w8)) and the baseline visit (week 0 (w0)) and was calculated with the following formula: ((w8 − w0)/w0) × 100. Due to the proof-of-concept nature of this study, raw values were also used for between-group comparisons after the treatment period at 8 w. This was carried out to indicate potential subtle treatment effects that may be further addressed in future studies; however, the analysis was only performed for variables in which there were no significant differences between the two groups at baseline. Independent data were analyzed by independent *t*-tests and Mann-Whitney U-tests. Normality testing was carried out with Shapiro-Wilk test. All statistical tests were performed using GraphPad Prism 9 (GraphPad Software Incorporated, La Jolla, CA, USA) or IMB SPSS statistics 27 (IBM Corp., Armonk, NY, USA). Significant differences were highlighted in tables. Data was considered significant if *p* < 0.05. As the study was an initial investigation, no adjustments were made for multiple comparisons.

#### 2.8.3. Compliance

A compliance table was designed and provided to the participants at the beginning of the intervention. Participants were instructed on the importance of taking the study products and complying with the study procedures. In addition to the compliance chart, the capsule containers were collected at the end of the study, and remaining capsules were counted as an additional measure of compliance. In the elderly cohort, all participants met compliance criteria (consumed study products at least 85% of the days during the eight weeks and consumed study products five out of seven days before a study visit), except one placebo participant. All data for this participant was removed from analysis.

## 3. Results

### 3.1. Sample Characteristics

Of 91 randomized participants, 76 (27 male) of median age 71 years (IQR 61–85) completed the study and were included in the analysis (Figure 1). In total, 14 participants declined participation, and one participant failed to meet compliance. As half of the study was conducted during the onset of COVID-19 (early 2020), half of the participants discontinued participation due to concerns with the study visits taking place at the university hospital. Two discontinued due to adverse events, one due to illness, and four discontinued for unspecified reasons. The baseline characteristics were similar between groups as shown in Table 1.

### 3.2. Effect of Dual Supplementation on hs-CRP Levels and Other Inflammatory Markers

The primary objective of the study was to assess the effect of the study supplements on low-grade inflammation as assessed by levels of the inflammatory marker hs-CRP. Levels of hs-CRP before and after the intervention are shown in Table 2. There was no significant difference between the active treatment group (median: 5.88%; IQR: −41.36–47.90) and the placebo (median: 3.52%; IQR: −15.53–55.66). However, comparison of hs-CRP levels after the eight-week intervention revealed a trend of lower values of hs-CRP in the active treatment group (median: 1.49; IQR: 1.15–2.16) compared to placebo (median: 1.79; IQR: 1.06–3.75) (*p* = 0.095).

In addition to hs-CRP, several other immune markers were examined to assess the effects of the supplements on immune function (Table 2). Levels of the anti-inflammatory cytokine IL-10 in the active treatment group (median 0.26 pg/mL; IQR: 0.16–0.36) were significantly higher than the placebo group (median 0.24 pg/mL; IQR: 0.11–0.29) (*p* = 0.05) after the study period, w8. There were no significant differences in IL-12p70, IL-6, TNF-α, IFN-γ, IL-8, IL-10, and IL-4 levels between the active treatment and placebo groups.

### 3.3. Gut-Health

#### 3.3.1. Effects on Intestinal Permeability

In vivo gastrointestinal permeability

The supplementation did not result in any differences in gastrointestinal permeability between the groups, as assessed by the multi-sugar urinary recovery test (Table 3).

##### I-FABP

I-FABP is a cytosolic protein present in differentiated enterocytes of the small intestine and to a lesser extent in the colon. In normal conditions, this protein is present in low amounts in the circulation, but upon damage to the intestinal barrier, it is released into the bloodstream. Increased I-FABP levels have been demonstrated in intestinal ischemia [39], celiac disease [40], necrotizing enterocolitis [41], and is hence a potential marker of intestinal injury and increased barrier permeability.

There was no significant difference in plasma intestinal fatty acid-binding protein (I-FABP) between the intervention and the placebo group—neither at any of the measured time points nor in % change between groups, refer to Table 3.

##### SCFA in Stool

Out of the nine measured SCFA acids (acetic, propionic, iso-butyric, butyric, isol-valeric, valeric, iso-carpoic, haxanoic and heptanoic acid) only one turned out significantly different between the two intervention groups: valeric acid was significantly increased in the treatment group (median: 3.7, IQR: 2.4–4.8) as compared to placebo (median: 2.5, IQR: 1.8–3.5) at the end of the intervention (*p* < 0.05). In addition, there was a trend of an increase in the iso-butyric acid in the treatment group, refer to *p*-value for % change in Table 3.

### 3.4. Mobility

The supplements had no significant effects on mobility measures. Refer to Appendix A for more detailed results.

### 3.5. Health Monitoring

No statistical change was observed between the groups for either BMI, insulin or glucose levels. Insulin and glucose were used to calculate the HOMA-IR. Baseline corrected values in placebo vs. treatment group for BMI was −0.04 (−0.95–0.57) and 0.04 (−0.78–0.78), respectively, and for HOMA-IR 0.06 (−3.16–1.49) vs. 0.02 (−2.50–1.66).

Any adverse events and adverse effects were recorded by the study staff. The diagnosis, if available, or each sign and symptom were recorded, and the Primary Investigator assessed each event with regard to its severity. The severity of the adverse events was determined according to the following classification:-Mild: Transient symptoms, no interference with the subject’s daily activities-Moderate: Marked symptoms, moderate interference with the subject’s daily activities but still acceptable-Severe: Considerable interference with the subject’s daily activities, unacceptable.

19 minor adverse events were recorded, which included change in stool frequency and consistency, joint and muscle pain, headache and nausea. 9/19 of the adverse events were in the active treatment group and 10/19 were in the placebo group; therefore, there was an even distribution in both groups. One placebo participant had a moderate adverse reaction. The frequency and severity of the adverse events were not higher than expected for a nutritional supplement intervention.

### 3.6. Compliance

In order to validate compliance and to assess the efficacy of the intervention, EPA, DHA, *n*-3 PUFA, *n*-6, and vitamin D levels were assessed at baseline and after the intervention Table 4. The active supplements resulted in a significant increase in vitamin D levels (median: 31.55, IQR: 27.20–37.15) compared to the placebo (median: 28.85; IQR: 25.43–32.73) (*p* = 0.018). The percent change in EPA (*p* = 0.0001), DHA (*p* = 0.0003), EPA + DHA (*p* = 0.0001), and *n*-3 PUFA (*p* = 0.0001) levels were significantly higher in the active treatment group compared to placebo. Furthermore, eight-week supplementation with the active products resulted in a 40.67% decrease (IQR: −61.36–20.74) in the *n*-6:*n*-3 ratio compared to a 1.37% increase (IQR: −19.20–13.18) in the placebo group. 

## 4. Discussion

Here, we present the first study to investigate the dual effects of a multi-strain probiotic and omega-3 PUFA supplements on low-grade inflammation in elderly participants. As the *Lactobacilli* and *Bifidobacteria* strains in the multi-strain probiotic [12,13,14,15] and *n*-3 PUFA supplements [16,24] have been shown to reduce inflammation, we hypothesized that the synergy of these supplements may provide additive health benefits. The study did not find any significant treatment effects, although a trend of lower values of hs-CRP, a well-characterized marker of inflammation, was seen in the active treatment group compared to placebo at 8 w. These findings are consistent with another study that could not observe a positive effect of a multi-strain probiotic (containing *B. longum* Bar33 and *L. helveticus* Bar 13) on hs-CRP levels in elderly participants [42]. On the other hand, a study that investigated the effects of a soluble corn fiber prebiotic in combination with *Lactobaillus rhamnosus* GG-PB12 found that this combination of supplements resulted in significantly decreased CRP levels compared to baseline in healthy elderly [13]. Furthermore, a meta-analysis showed that both probiotics and omega-3 supplements are efficacious in reducing inflammation as measured by IL-6 and CRP [25]. The lack of significant hs-CRP decreases in the current study may suggest that not all probiotic strains are efficacious in reducing low-grade inflammation in the elderly. Furthermore, a higher dose of omega-3 may have also been necessary to see positive effects. The lack of significant results might also be related to under-powering issues in the current study. Finally, the study was conducted during the COVID-19 pandemic; therefore, it is possible that the changes in stress, activity, socialization, etc. may have impacted the results of the study.

An additional factor that may explain the failure of our intervention to reduce hs-CRP levels is the variability of hs-CRP as a marker of low-grade inflammation. In our study, we observed considerable variability in hs-CRP levels from the screening to the baseline visit, even though these visits were at most two to three weeks apart. Therefore, several of the participants that were included as having low-grade inflammation from the screening visit did not meet the inclusion criteria at baseline. Our findings are consistent with several other studies that observe significant short-term within-person variability in CRP levels [43,44,45]. The intra-individual variability of hs-CRP observed in these studies as well as the present study has implications for its utility as a marker of low-grade inflammation. These findings highlight the necessity of identifying more stable markers for studies that investigate changes in low-grade inflammation or that several hs-CRP screening measurements need to be taken to reduce intra-individual variation [46].

Although we failed to see a significant reduction in hs-CRP, we observed a trend toward higher levels of the anti-inflammatory cytokine IL-10 after the intervention in the active treatment group compared to the placebo. Even though the findings are quite modest, they suggest that the probiotic omega-3 supplementation may be effective in increasing anti-inflammatory cytokine production. This finding is consistent with several other probiotic interventions in healthy elderly that observed significant increases in IL-10 [47,48]. Furthermore, findings from a study in mice suggest that omega-3 may affect inflammation via increasing IL-10 expression [49]. IL-10 plays a significant role in reducing inflammation by suppressing the activation of macrophages and suppressing the release and activity of inflammatory cytokines such as IL-6, IL-8, and TNF-α [50,51]. Levels of IL-10 have been shown to decrease with aging, and age-related dysregulation of IL-10 production has been implicated in inflammaging and other health complications [52,53,54]. Therefore, by increasing levels of IL-10, the dual supplementation utilized in this intervention may have potential in reducing some of the age-related decline in anti-inflammatory cytokines.

To further evaluate the impact of the study products on inflammation, other measurements in addition to levels of cytokines could be utilized. The majority of probiotic interventions in healthy elderly observed no or modest effects on serum markers [15]. However, several studies examined the effects of probiotic interventions on natural killer (NK) cell activity. For example, an intervention with *B. lactis* (HN019) significantly increased the phagocytic capacity of the NK cells compared to placebo [55]. In addition to examining NK cell activity, other studies characterized B and T cell populations. Supplementation with *B. longum* Bar33 and *L. helveticus* Bar13 significantly increased the percentages of CD4+ and CD8+ naive T cells and decreased the number of CD4+ effector memory T cells [42]. Similarly, a multi-strain probiotic consisting of *L. delbrueckii* spp. bulgaricus, 8481 and *S. thermophilus*, 8357 significantly decreased CD8+ lymphocytes and increased the number of NK cells [12]. As hs-CRP and cytokines can be subject to intra-individual variability and have been shown to be rather modestly impacted by probiotic and synbiotic interventions, investigating immune cell activity as well as immune cell phenotypes could be a more useful method to accurately assess the effects of nutritional supplements on inflammation.

Concerning the gut-related measures included in this trial, most were unaffected by the intervention and also did not differ between the two groups. The only potentially interesting difference found was in valeric acid levels, which were significantly increased in the treatment group compared to placebo at eight weeks. SCFAs have gained significant attention in recent years, and, to our knowledge, the focus has primarily been on their health-promoting properties. These fatty acids are metabolites primarily produced by fermentation of undigested dietary carbohydrates in the colon. SCFAs, particularly butyrate, exert multiple effects on human health such as the inhibition of colonic carcinogenesis, inflammation, and oxidative stress, as well as improvement of colonic barrier function [56]. Hence, SCFAs are metabolites that play significant roles in both immune regulation and intestinal barrier function and are arguably of interest for the current study. In the current trial, we found no effects on butyrate; however, the tested supplements had a significant effect on valeric acid—another SCFA that is less commonly considered in the literature. Valeric acid is longer than butyrate and originates in 5-aminovalerate, which is a product of the anaerobic degradation of previously hydrolyzed protein by gut bacteria [57]. The scarce scientific literature on valeric acid is somewhat contradictory in its purposed effects on human and mammalian health. A pilot study in constipated patients showed a reduction in symptom severity accompanied by increased levels of fecal valeric acid [58]. On the other hand, an additional study demonstrated that surgical intervention elicits increases in plasma valeric acid in mice, which may be further coupled to learning impairments and detrimental effects in the mouse brain [59]. In addition, ischemic stroke is associated with increased concentrations of valeric acid, and changes in valeric acid have also been positively correlated with hs-CRP levels and white blood cell counts in these patients [60]. Moreover, in this study, the increase in valeric acid was accompanied by an increase in IL-10. In support of this finding, a few previous studies have observed simultaneous increases of valeric acid and IL-10. For example, a rat model utilizing a lipopolysaccharide challenge to trigger inflammation showed that Yanning Syrup (a commercially sold and patented Chinese medicinal product) up-regulated anti-inflammatory cytokines (including IL-10) while increasing SCFA acids (including valeric acid) [61]. Nicotinamide could restore valeric acid and IL-10 levels (i.e., increasing them) in a mouse colitis model [62]. Furthermore, Fucoidan (a long chain sulfated polysaccharide found in brown algae) increased IL-10 and valeric acid in a rat model of induced colorectal cancer [63]. In addition, in humans, fried meat consumption reduces levels of fecal SCFAs (including valeric acid) and attenuates plasma cytokine levels (including among IL-10), while also producing significant and adverse changes in the gut microbiota of obese adults [64]. Taken together these findings point towards valeric acid as possibly being involved in favorable immune regulation. However, these findings are merely simultaneous observations, and the causation is not thoroughly established. Hence, the current literature paints an inconclusive picture concerning the effects of valeric acid on mammalian health and immune regulation, and further studies are needed to clarify this relationship.

## 5. Conclusions

Here, we present the first study to investigate the dual effects of a multi-strain probiotic and omega-3 supplements on low-grade inflammation in elderly participants. It is important to note, however, that the current study was designed as a proof-of-concept study to initially assess the efficacy of dual supplementation to reduce inflammation in older adults. To provide further insight, this small-scale trial would need to be followed by a larger trial with an arm for each of the supplements alone and in combination. The eight-week dual supplementation, as investigated in the current study, had no significant effects on the primary outcome of hs-CRP but showed some modest post-intervention differences in the anti-inflammatory marker IL-10 and levels of valeric acid in stool samples. We consider these findings to be in line with the underlying hypothesis of these supplements being able to reduce inflammaging in older adults. However, we want to stress that the interpretation relies on weak indications from this particular study and would thus have to be further supported in future studies. However, taken together with findings from similar previous studies, our results suggest that probiotic and omega-3 dual supplementation exerts modest effects on inflammation and may therefore be interesting to pursue as a non-pharmacological treatment for low-grade inflammation in elderly in future studies.

## Figures and Tables

**Figure 1 nutrients-14-03998-f001:**
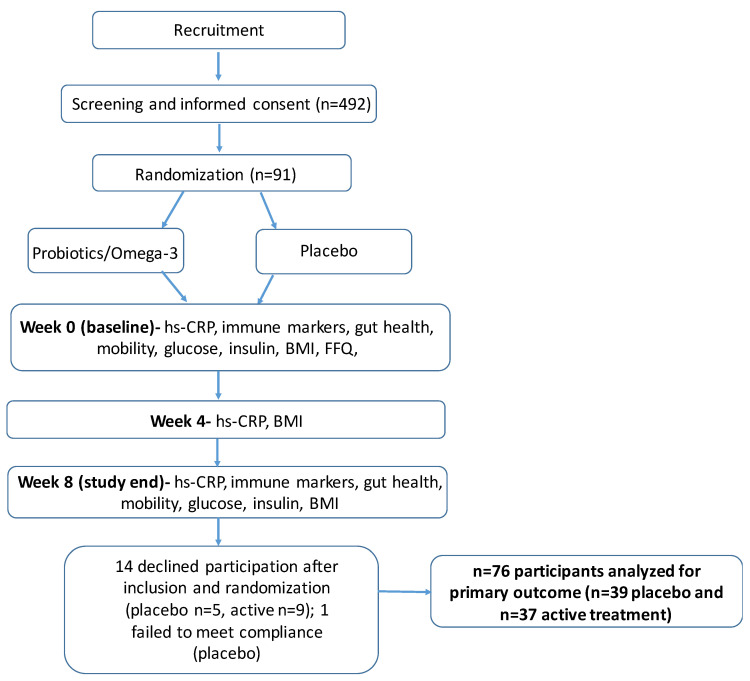
Study design.

**Table 1 nutrients-14-03998-t001:** Participant characteristics at baseline.

	Placebo (*n* = 39)		Treatment (*n* = 37)		*p*-Value
women:men [%]	64.1	35.9	64.9	35.1	
count	25	14	24	13	
age [years]	70.0	65–81	71.0	65–80	0.261
height [cm]	169.2	158–186	169.5	151–186	0.925
weight [kg]	72.2	47–91	70.6	50–92	0.938
BMI [kg/m^2^]	24.6	18–27	24.7	20–27	0.767
Fiber [g]	25.7	4.6–133.5	24.6	11.0–95.9	0.618
EPA [g]	0.10	0.01–0.34	0.10	0.01–0.41	0.912
DHA [g]	0.21	0.02–0.65	0.20	0.01–0.85	0.679
Vitamin D [µg]	8.05	2.48–34.2	7.35	3.43–16.1	0.791

Values are given as medians (IQR). Values for fiber, EPA, DHA, and vitamin D were derived from the FFQ at baseline.

**Table 2 nutrients-14-03998-t002:** Immune-related measurements.

		Placebo				Active Treatment			Significance	
	*n*	Pre-Treatment	Post- Treatment	% Change Pre-Post	*n*	Pre- Treatment	Post- Treatment	% Change Pre-Post	*p*: Post-Treatment	*p*: % Change Pre-Post
hs-CRP [g/mL]	39	1.77 (1.04–3.240)	1.79 (1.06–3.75)	5.56 (−70.53–32.39)	37	1.43 (0.94–1.89)	1.49 (1.15–2.16)	3.40 (−18.41–35.76)	0.095 *	0.50
IL-12p70 [pg/mL]	37	0.069 (0.038–0.11)	0.074 (0.044–0.12)	0.064 (−25.52–63.47)	35	0.065 (0.041–0.086)	0.075 (0.041–0.092)	−0.74 (−15.76–13.67)	0.55	0.39
IL-6 [pg/mL]	39	0.51 (0.34–1.23)	0.65 (0.42–1.06)	10.15 (−16.76–55.51)	37	0.44 (0.35–0.80)	0.53 (0.40–0.84)	8.18 (−18.97–36.61)	0.16	0.36
IFN-gamma [pg/mL]	39	5.53 (3.34–8.95)	5.68 (3.23–10.30)	0.31 (−29.66–40.61)	37	5.87 (3.84–7.88)	6.16 (3.22–8.65)	−2.65 (−28.03–33.87)	0.87	0.94
TNF [pg/mL]	39	1.25 (0.93–1.98)	1.24 (0.89–1.93)	−1.69 (−14.43–13.11)	37	1.19 (0.66–0.11)	1.14 (0.83–1.52)	−3.05 (−10.02–5.47)	0.65	0.80
IL-8 [pg/mL]	39	13.51 (11.30–16.83)	12.87 (10.42–171.2)	−2.66 (−16.58–12.74)	37	12.42 (9.48–16.25)	11.53 (9.35–17.57)	−5.35 (−17.93–8.69)	0.50	0.66
IL-10 [pg/mL]	39	0.21 (0.12–0.27)	0.24 (0.11–0.29)	14.79 (−9.96–32.81)	37	0.23 (0.14–0.34)	0.26 (0.16–0.36)	18.23 (−0.38–32.71)	0.050 **	0.54
IL-4 [pg/mL]	39	0.019 (0.013–0.028)	0.015 (0.012–0.026)	−10.91 (−26.67–22.57)	37	0.016 (0.0091–0.026)	0.014 (0.0085–0.020)	−27.23 (−45.72–52.56)	0.38	0.42

Data displayed as median (IQR). * indicates *p* < 0.1 and ** indicates *p* <0.05

**Table 3 nutrients-14-03998-t003:** Gut-health related measurements.

		Placebo				Active Treatment			Significance	
	*n*	Pre-Treatment	Post- Treatment	% Change Pre-Post	*n*	Pre- Treatment	Post- Treatment	% Change Pre-Post	*p*: Post-Treatment	*p*: % Change Pre-Post
In vivo intestinal permeability										
Sugar excretion:										
0–5 h:										
L/R ratio	37	0.03 (0.02–0.06)	0.04 (0.02–0.06)	13.2 (−11–59.7)	34	0.03 (0.01–0.04)	0.03 (0.02–0.05)	4.2 (−25.9–44.5)	0.27	0.33
Sucrose [μg/mL]	38	55.5 (13.8–181.6)	41.8 (12.8–381.1)	0.05 (−40.82–46.4)	34	41.1 (17.8–188.3)	36.2 (18.1–275.7)	−4.9 (−39.8–29.4)	0.92	0.58
6–24 h:										
S/E ratio	35	0.03 (0.02–0.03)	0.02 (0.02–0.03)	−2.8 (−27.1–24.4)	31	0.02 (0.02–0.03)	0.02 (0.02–0.03)	−1.4 (−20.8–22.2)	0.31	0.98
Intestinal damage marker										
I-FABP [pg/mL]	39	382.8 (264–545.6)	329 (256.4–565.5)	−6 (−34.3–55.8)	37	352.3 (242.6–621.7)	381.2 (218.3–608.6)	−0.4 (−29.6–40.4)	0.56	0.94
Stool short-chain fatty-acids										
[µmol/g in fecal dry mass]										
Acetic acid	19	87.1 (72.7–113.8)	89.3 (75.7–120)	4.1 (−18.7–29.8)	20	95.2 (74.1–140.5)	97.6 (79.4–143.6)	−6.4 (−27.6–51.3)	0.55	0.81
Propionic acid	19	18.7 (15.2–24.4)	20.5 (12.8–25.3)	7.2 (−37.9–40)	20	17.1 (12.2–45.8)	18.1 (14.6–33.7)	−2.5 (−40.1–75.8)	0.99	0.86
Iso-butyric acid	19	2.9 (2.4–3.8)	2.9 (2–3.8)	−9 (−30.4–31.5)	20	2.3 (1.8–2.9)	3.4 (2.2–5)	24.7 (−26.6–130.5)	0.32	0.0948 *
Butyric acid	19	16.2 (8.7–24.9)	14.9 (7.6–26.1)	24.2 (−50.3–95)	20	14.9 (9.2–24.3)	15.8 (11.83–29.8)	19.8 (−44.3–131.4)	0.53	0.61
Iso-valeric acid	19	4.1 (3.1–5.5)	4.3 (3–5.4)	−6.9 (−19.2–38.9)	20	3.4 (2.3–4.5)	5.1 (3–7.1)	20.5 (−22.7–100.5)	0.19	0.18
Valeric acid	19	2.3 (1.9–3.5)	2.5 (1.8–3.5)	2.9 (−31.2–48.6)	20	2.9 (2.1–5.1)	3.7 (2.4–4.8)	10.9 (−36.7–94.4)	0.0436 **	0.56
Iso-caproic acid	19	0.2 (0.1–0.5)	0.2 (0.2–0.5)	12.8 (−49.6–117.8)	20	0.4 (0.2–0.5)	0.4 (0.3–0.6)	25.1 (−21.5–198.6)	0.20	0.73
Hexanoic acid	19	0.5 (0.3–1.79	0.7 (0.4–2)	14.4 (−24.5–112.49	20	1.1 (0.6–1.7)	1.4 (0.8–2.4)	3.6 (−35.6–78.99)	0.24	0.86
Heptanoic acid	19	0.5 (0.4–0.6)	0.4 (0.3–0.5)	−11.9 (−68.6–27.5)	20	0.4 (0.2–0.5)	0.5 (0.3–0.69	7.8 (−25.9–170)	0.17	0.12

Data displayed as median (IQR). * indicates *p* < 0.1 and ** indicates *p* < 0.05

**Table 4 nutrients-14-03998-t004:** Validation of study compliance.

		Placebo				Active Treatment			Significance	
	*n*	Pre- Treatment	Post- Treatment	% Change Pre-Post	*n*	Pre- Treatment	Post- Treatment	% Change Pre-Post	*p*: Post-Treatment	*p*: % Change Pre-Post
Vitamin D [ng/mL]	36	25.00 (19.78–28.28)	28.85 (25.43–32.73)	16.04 (9.44–32.62)	34	24.50 (20.90–30.53)	31.55 (27.20–37.15)	23.98 (11.75–38.43)	0.018 **	0.12
EPA [mg/100 mL]	39	4.64 (3.27–5.87)	3.79 (2.21–5.68)	−13.64 (−37.57–17.45)	37	3.53 (2.61–5.95)	7.56 (5.13–9.64)	46.23 (24.82–61.93)	0.0001 **	0.0001 **
DHA [mg/100 mL plasma]	39	13.29 (10.14–14.79)	11.64 (4.48–15.63)	−20.20 (−77.20–5.62)	37	9.48 (5.91–13.37)	12.16 (8.17–14.47)	13.80 (−13.58–36.25)	Baseline difference	0.0003 **
EPA + DHA [mg/100 mL]	39	17.60 (14.42–20.92)	14.92 (6.29–21.35)	−21.64 (−74.06–6.31)	37	13.86 (9.12–23.76)	19.68 (12.85–23.76)	28.69 (3.91–44.22)	Baseline difference	0.0001 **
*n*-3 PUFA [mg/100 mL]	39	23.55 (20.02–26.75)	19.76 (9.07–29.19)	−19.03 (−79.45–8.22)	37	18.42 (11.57–23.98)	24.82 (17.27–28.38)	28.50 (−2.78–41.06)	Baseline difference	0.0001 **
*n*-6:*n*-3 ratio [mg/100 mL]	39	4.89 (4.13–5.48)	4.66 (4.11–5.91)	1.37 (−19.20–13.18)	37	4.40 (3.85–5.11)	3.23 (2.81–3.75)	−40.67 (−61.36–20.74)	0.0001 **	0.0001 **

Data displayed as median (IQR). ** indicates *p* < 0.05

## Data Availability

Not applicable.

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
