# Peer review of "Potential Modulation of Inflammation by Probiotic and Omega-3 Supplementation in Elderly with Chronic Low-Grade Inflammation—A Randomized, Placebo-Controlled Trial"

_nutrients, 2022, doi:10.3390/nu14193998_

Round 1

Reviewer 1 Report

The main aim of this manuscript was to assess if the combination of a multi-strain probiotic enriched with n-3 PUFA will decrease low-grade systemic inflammation in elderly participants. Assessing the benefits of probiotics for the elderly group is an important and current issue and there is still a lack of research in this area.

In my opinion, the introduction omits important papers examining the potential effect of probiotics on inflammatory biomarkers, which might be for example related to the underlying mechanisms in cognitive disorders (Den et al. 2020).

Moreover, the authors do not mention the paper which compared to the potential effect of different compounds to reducing biomarkers of systemic inflammation in clinical trials conducted to  older adults with chronic low-grade inflammation (Custodero et al. 2018). In this study the effect of probiotics was significantly larger compared to those of omega-3. But it is still unknown whether dual supplementation will differ from a single supplementation.

I consider the selection of articles for the paragraph on dual supplementation to be limited, and suggest elaborating on this important part.

The authors emphasize many times in the paper that synergy of these supplements may provide an additive health benefit than when either supplement is administered alone, but the study did not include the comparative group (with single supplementation) to check the differences between dual supplementation and when supplement is administered alone, which in my opinion is a great limitation of this study.

The primary outcome, was also to explored effects of the supplements on physical function, but these outcomes were assessed only to a limited extent (based on an self-reported questionnaire and one test). I encourage you to consider omitting this element in order to slightly shorten a very long manuscript.

I also have some comments on tables and figures. It would be beneficial to describe the measured variables in more detail in Figure 1. For example, what does "overall health assessment" mean? At what point does the participant fills up the FFQ? etc. Table 1 shows data on fiber, DHA, EPA, vitamin D based on FFQ? It is not clear. This is also not explained in the methodology section. 

Author Response

  1. The main aim of this manuscript was to assess if the combination of a multi-strain probiotic enriched with n-3 PUFA will decrease low-grade systemic inflammation in elderly participants. Assessing the benefits of probiotics for the elderly group is an important and current issue and there is still a lack of research in this area. We thank the reviewer for the insightful comments. We are pleased to hear that the reviewer agrees that the current study is of interest to the field.
  2. In my opinion, the introduction omits important papers examining the potential effect of probiotics on inflammatory biomarkers, which might be for example related to the underlying mechanisms in cognitive disorders (Den et al. 2020). We agree with the reviewer that this is quite interesting, and a sentence including this reference has been added to the Introduction. 
  3. Moreover, the authors do not mention the paper which compared to the potential effect of different compounds to reducing biomarkers of systemic inflammation in clinical trials conducted to older adults with chronic low-grade inflammation (Custodero et al. 2018). In this study the effect of probiotics was significantly larger compared to those of omega-3. But it is still unknown whether dual supplementation will differ from a single supplementation. We thank the reviewer for suggesting that we include this systematic review.  It has now been added to the Dual supplementation section in the Introduction. 
  4. I consider the selection of articles for the paragraph on dual supplementation to be limited, and suggest elaborating on this important part. We completely agree with this suggestion, and we have now expanded this section.  We have now included the systematic review as you suggested in comment #3.  We have also added a study that demonstrates that omega-3 and probiotics administered together have more pronounced effects on inflammation than either supplement administered alone (Rajkumar 2014).  Finally, we highlighted the potential of omega-3 to act as a prebiotic, which may be a mechanism underlying potential synergistic effects. 
  5. The authors emphasize many times in the paper that synergy of these supplements may provide an additive health benefit than when either supplement is administered alone, but the study did not include the comparative group (with single supplementation) to check the differences between dual supplementation and when supplement is administered alone, which in my opinion is a great limitation of this study. We agree that this is a limitation of the current study.  As we are interested in the efficacy of dual supplementation to affect inflammation, we have removed several of the words “than when either supplement is administered alone” to be more clear that this comparison was not an objective of the current study.  In addition, we have added to the Conclusion that a future studies with additional arms with each supplement administered alone is crucial. 
  6. The primary outcome, was also to explored effects of the supplements on physical function, but these outcomes were assessed only to a limited extent (based on an self-reported questionnaire and one test). I encourage you to consider omitting this element in order to slightly shorten a very long manuscript. We agree with the reviewer’s insight that the manuscript is quite long. We think it is important to include the physical function measures as they were a secondary outcome in our protocol; however, we agree to move the results for this outcome to supplementary material.  We also removed the section about the mobility measurements from the Discussion. 
  7. I also have some comments on tables and figures. It would be beneficial to describe the measured variables in more detail in Figure 1. For example, what does "overall health assessment" mean? At what point does the participant fills up the FFQ? etc. Table 1 shows data on fiber, DHA, EPA, vitamin D based on FFQ? It is not clear. This is also not explained in the methodology section.  Thank you for this suggestion. We have now altered Figure 1 to include more information about the measured variables, including specifying that the FFQ is filled out at baseline.  In our methods section, we have noted that the FFQ occurs at the baseline visit, and this has also now been clarified in a more detailed Figure 1.  In addition, we clarified in the text for Table 1 that the fiber, DHA, EPA, and Vitamin D estimates came from the FFQ at baseline. 

Reviewer 2 Report

This is a nicely designed study that tested the hypothesis that in healthy older adults a probiotic in combination with fish oil reduces chronic inflammation as measured by CRP. Secondary outcomes were numerous and diverse.

Primary concerns and recommendations

1.       As an RCT with baseline and post-intervention methods, all results should be analyzed as change scores. Ideally, all analyses would also be adjusted for baseline measures.  This can be achieved with repeated measures ANOVA or linear regression. The Methods section lacks an analytic plan suitable for a RCT. The authors generally table the findings nicely by reporting baseline, post-intervention and change scores. However, I believe the manuscript is inconsistent how the outcomes are analyzed, often not presenting change score analysis and not generally adjusting for baseline. The figures also are inconsistent in this regard. This is particularly problematic with the analysis and presentation of CRP and IL-10.

2.       Since the study examines the combined effects of a probiotic and omega-3 fatty acids and would like to clarify whether there are additive or synergistic effects, the lack of probiotic only and omega-3 only groups is a serious weakness. This should be explained and discussed in the Introduction, Methods and Discussion.

3.       The Methods should explain the specific rationale for the chosen probiotic and the dose of fish oil. (It was rather small and, I suspect, smaller than trials upon which this study is based. Why was stool microbiota not an outcome?  The inclusion of vitamin D in the fish oil is problematic since it adds another active ingredient to the intervention arm. Are these weaknesses that warrant more attention in the Discussion?

4.       The measure of intestinal permeability is new to me. From the information provided, I cannot understand why the urine collection was separated by hours 0-5 and 6-24. In addition, I do not understand how the data were analyzed. Sucrose is listed as a non-metabolizable sugar, but I believe that is not true. Also, the presented data on sucrose in Table 3 seems to be in error for the active treatment arm: pre-treatment 52.1, post-treatment 50.75, but % change given as 23.5%(!).

5.       The flow chart is good, but the 14 drop-outs require more explanation. Apparently, all enrolled participants gave informed consent and underwent screening procedures to determine eligibility prior to randomization. Therefore, why then did these 14 subjects “decline participation” after randomization? If they dropped out at different steps of the trial, please indicate at roughly what point and for what reason each subject dropped out (e.g., side effects, moved from the area, compliance with study procedures, etc).

6.       The baseline CRP data in Table 1 indicate a maximum of 18.2 in the intervention group compared to just 3.38 in the control group. I strongly suspect this is due to at least one high outlier in the intervention group. I also suspect that distribution of CRP is highly skewed. How were these distribution issues handled? Such a high baseline CRP strongly suggests an acute inflammatory state which then clearly confounds the interpretation of treatment effects. Perhaps the authors can consider excluding subjects with a high baseline CRP (as is often done in these studies).  

7.       The study had many, many outcomes and, therefore, has a high risk of type 1 statistical error. The usual approach is the adjust all secondary analyses to for multiple comparisons. The alternative is repeatedly caution readers that these findings are exploratory and could well be due to chance.

Secondary comments and recommendations

8.       The introduction could be improved by not simply summarizing that probiotics and omega-3 fatty acids seem to reduce inflammation but describe whether and how you posit that their metabolic and physiologic effects would interact. Otherwise, there is not really a scientific rationale to test their “interaction.”

9.       The authors do a good job of reviewing and citing relevant literatures. However, there is a recent systematic review of the effects of omega-3 fatty acids on sarcopenia they may wish to include -- The effect of long chain omega-3 polyunsaturated fatty acids on muscle mass and function in sarcopenia: A scoping systematic review and meta-analysis. Bird JK, Troesch B, Warnke I, Calder PC.  Clinical Nutrition ESPEN. 46:73-86, 2021 12.

10.   The Methods includes more detail than is typical. Some could be omitted by either referencing prior reports or placement an online supplement.

11.   The rationale for measuring short-chain fatty acids in stool is not clear to me.

12.   The fasting insulin and glucose values are typically used to calculate HOMA-IR, which is more informative with respect to insulin resistance than either insulin or glucose alone. Perhaps the authors could report treatment effects on HOMA-IR.

13.   Compliancy should be changed to compliance.

14.   The Discussion considers the putative effects of probiotics on the trial outcomes in great detail without much of any consideration about the putative effects of omega-3 fatty acids.

15.   I do not believe the authors can make any statements about non-significant effects on any outcomes (i.e., trends in the hypothesized direction).  Doing so in Discussion sections generally reflect author bias. In addition, I suspect some of these favorable trends may not persist after consideration of the statistical issues noted above. So, I would ask the authors to generally remove such statements from the Discussion.

16.   The Discussion, like the Methods, is too long. Would reduce by 30%.

17.   There are typographical errors in several places. Careful proofreading by the authors is needed.  

Author Response

This is a nicely designed study that tested the hypothesis that in healthy older adults a probiotic in combination with fish oil reduces chronic inflammation as measured by CRP. Secondary outcomes were numerous and diverse. Thank you so much for reviewing our manuscript. We are pleased to hear that the reviewer finds our study to be nicely designed. 

Primary concerns and recommendations

  1. As an RCT with baseline and post-intervention methods, all results should be analyzed as change scores. Ideally, all analyses would also be adjusted for baseline measures.  This can be achieved with repeated measures ANOVA or linear regression. The Methods section lacks an analytic plan suitable for a RCT. The authors generally table the findings nicely by reporting baseline, post-intervention and change scores. However, I believe the manuscript is inconsistent how the outcomes are analyzed, often not presenting change score analysis and not generally adjusting for baseline. The figures also are inconsistent in this regard. This is particularly problematic with the analysis and presentation of CRP and IL-10. We thank the reviewer for pointing this out, and we have now made an effort to improve our Data Analysis section, clarifying our analytical plan. The main statistical strategy is using percent change (between baseline and 8 weeks) as the basis for our analyses. We calculate this by entering the values for the study end visit (week eight: w8) and the baseline visit (week zero: w0) into the following formula: [(((w8-w0)/w0)*100)]. This creates a baseline correction for our variables. However, we additionally reasoned that comparing the after-treatment effects by “raw values”, for variables where there were no significant differences in baseline scores between the groups, may be interesting and thus included this as well. The analysis is thus consistent with our study protocol approved by the ethical review board. Finally, we decided to highlight any significant findings by figures and leave all the other data displayed in tables, now clearly stated in the Data Analysis section.
  2. Since the study examines the combined effects of a probiotic and omega-3 fatty acids and would like to clarify whether there are additive or synergistic effects, the lack of probiotic only and omega-3 only groups is a serious weakness. This should be explained and discussed in the Introduction, Methods and Discussion. We realize that we have not clearly stated that the current study is a proof-of-concept study, and we want to express our gratitude to the reviewer for making us aware of this. Proof-of-concept studies are smaller trials that are meant to determine the feasibility of the idea or to verify that the idea will function as envisioned. Here, we were specifically interested in the potential efficacy of combining these supplements. We have now clarified this in the Introduction, Methods, and Discussion/Conclusion of the manuscript and hope this is more to the liking of the reviewer.
  3. The Methods should explain the specific rationale for the chosen probiotic and the dose of fish oil. (It was rather small and, I suspect, smaller than trials upon which this study is based. Why was stool microbiota not an outcome?  The inclusion of vitamin D in the fish oil is problematic since it adds another active ingredient to the intervention arm. Are these weaknesses that warrant more attention in the Discussion? The omega-3 supplements were provided by GSK/Pfizer; therefore, we were not able to alter the dose.  However, we agree that a higher dose may have elicited more beneficial effects, and we have now noted this in the Discussion.  Stool microbiota was an exploratory outcome, and we are considering to analyze these samples for a future manuscript.  We feel that the inclusion of vitamin D in the fish oil is not too problematic, as the dose is rather low.  However, we agree that this should warrant more attention in planning future studies.  
  4. The measure of intestinal permeability is new to me. From the information provided, I cannot understand why the urine collection was separated by hours 0-5 and 6-24. In addition, I do not understand how the data were analyzed. Sucrose is listed as a non-metabolizable sugar, but I believe that is not true. Also, the presented data on sucrose in Table 3 seems to be in error for the active treatment arm: pre-treatment 52.1, post-treatment 50.75, but % change given as 23.5%(!). We are happy to clarify this further.The sugar recovery ratios from an early urine fraction (urine collection of 0-5 hours after intake of sugars) and a later fraction (urine collection from 6-24 hours after sugar intake) are used to assess the intestinal permeability of the respective intestinal segments. This is possible because the intestinal segments of the upper gastrointestinal tract (evaluated in the 0-5 hour fraction) have shorter transit time and some sugars are absorbed earlier as compared to the lower segments (measured in the 6-24 hour fraction). We added some information that we hope clarifies this in the Methods section and also complemented with a few more references to support this. We have also added a reference in the method section that we hope will help with understanding the data analysis. Indeed, we recognize that the question about non-digestible sugars may be debated, and we hence removed the word non-digestible and hope that this resonates better with our potential readers. Considering Table 3, the displayed data looks strange because we have chosen to show only median values and not arithmetic means (we are doing so because the data is non-normally distributed and non-parametric tests have been used to analyze them). The median looks strange because there is an even number of participants in the treated group and, hence, the median becomes a mean of two numbers (the two most mid ones) which makes the numbers in the table look strange. But we have looked into this before and even re-analyzed this before we realized why the data looks as it does. We hope the reviewer will be satisfied with this response.
  1. The flow chart is good, but the 14 drop-outs require more explanation. Apparently, all enrolled participants gave informed consent and underwent screening procedures to determine eligibility prior to randomization. Therefore, why then did these 14 subjects “decline participation” after randomization? If they dropped out at different steps of the trial, please indicate at roughly what point and for what reason each subject dropped out (e.g., side effects, moved from the area, compliance with study procedures, etc). Our study was conducted in two cohorts- autumn 2019 and early spring 2020. The majority of our drop-outs were participants that were included and randomized but then decided to drop-out in the few days/weeks before study start as the Covid cases began to increase (Feburary 2020).  Two participants decided to drop-out because they decided they didn’t want to be included, two had issues with other medications/supplements, and one discontinued participation due to side-effects.  We have now included more information about the drop-outs in the beginning of the Results section. 
  2. The baseline CRP data in Table 1 indicate a maximum of 18.2 in the intervention group compared to just 3.38 in the control group. I strongly suspect this is due to at least one high outlier in the intervention group. I also suspect that distribution of CRP is highly skewed. How were these distribution issues handled? Such a high baseline CRP strongly suggests an acute inflammatory state which then clearly confounds the interpretation of treatment effects. Perhaps the authors can consider excluding subjects with a high baseline CRP (as is often done in these studies).  We highly appreciate the reviewer pointing out this error.  We had failed to adjust Table 1 for hs-CRP values that excluded this high value that you identified.  We have now used the correct adjusted data set, and the values now properly match what was shown for the baseline data in Table 2. 
  3. The study had many, many outcomes and, therefore, has a high risk of type 1 statistical error. The usual approach is the adjust all secondary analyses to for multiple comparisons. The alternative is repeatedly caution readers that these findings are exploratory and could well be due to chance. As the study was an exploratory, proof-of-concept study, we have included a wide range of outcomes.  We have taken the reviewer’s suggestion and indicated in several places in the Introduction, Discussion, and Conclusion that the study was exploratory and the findings need to be confirmed in future studies. 

Secondary comments and recommendations

  1. The introduction could be improved by not simply summarizing that probiotics and omega-3 fatty acids seem to reduce inflammation but describe whether and how you posit that their metabolic and physiologic effects would interact. Otherwise, there is not really a scientific rationale to test their “interaction.” As this is not a study of mechanisms per se but instead a proof-of-concept study to investigate whether this area would be suited for future in-depth studies, we have tried to keep our Introduction at a “depth” of information that we feel reasonable. In the Introduction, we make a case for both supplements to influence immune regulation as well as gut barrier function; from this evidence, we believe that they could act synergistically. We tried to clarify this further by adding the following sentences under Dual Supplementation section in the Introduction: “As outlined above, both probiotics and omega-3 fatty acids have been shown to have similar effects on immune regulation and intestinal barrier function. Yet, very few studies have thus far examined the effects of dual supplementation.”
  2. The authors do a good job of reviewing and citing relevant literatures. However, there is a recent systematic review of the effects of omega-3 fatty acids on sarcopenia they may wish to include -- The effect of long chain omega-3 polyunsaturated fatty acids on muscle mass and function in sarcopenia: A scoping systematic review and meta-analysis. Bird JK, Troesch B, Warnke I, Calder PC.  Clinical Nutrition ESPEN. 46:73-86, 2021 12. We appreciate the suggestion of this review and find it to be highly interesting; however, in light of suggestions from Reviewer 1, we have removed from the Discussion the paragraph about omega-3/probiotics and physical function.    
  3. The Methods includes more detail than is typical. Some could be omitted by either referencing prior reports or placement an online supplement. The Methods section is now shorter; details for the blood sample collection and analysis and short-chain fatty acids in stool samples have now been moved to supplementary material.  The protocol for the sample preparation and determination of sugar concentration is shorter and now references a previous manuscript. 
  4. The rationale for measuring short-chain fatty acids in stool is not clear to me. As we briefly outline in the Discussion (bottom of page 18 and top of page 19), SCFAs exert multiple effects on human health such as the inhibition of colonic carcinogenesis, inflammation, and oxidative stress, as well as improvement of colonic barrier function. Hence, SCFAs are metabolites that play significant roles in both immune regulation and intestinal barrier function and are arguably of interest for the current study. 
  5. The fasting insulin and glucose values are typically used to calculate HOMA-IR, which is more informative with respect to insulin resistance than either insulin or glucose alone. Perhaps the authors could report treatment effects on HOMA-IR. We decided to only include the raw values for glucose and insulin.  As there were no significant differences between groups, we decided not to include HOMA-IR values as well. 
  6. Compliancyshould be changed to compliance. Each “compliancy” has been changed to “compliance.”
  7. The Discussion considers the putative effects of probiotics on the trial outcomes in great detail without much of any consideration about the putative effects of omega-3 fatty acids. We have now expanded the Discussion in several places to further consider the putative effects of omega-3 fatty acids on the trial outcomes.   
  8. I do not believe the authors can make any statements about non-significant effects on any outcomes (i.e., trends in the hypothesized direction).  Doing so in Discussion sections generally reflect author bias. In addition, I suspect some of these favorable trends may not persist after consideration of the statistical issues noted above. So, I would ask the authors to generally remove such statements from the Discussion. We agree with the reviewer and have worked to minimize discussion about favourable trends.  
  9. The Discussion, like the Methods, is too long. Would reduce by 30%. We agree that the Discussion is too long and have removed two entire paragraphs as well as several sentences in other places. 
  10. There are typographical errors in several places. Careful proofreading by the authors is needed.  We thank the reviewer for noting this. More thorough proofreading has been done, and several errors are now corrected in the text.    

Round 2

Reviewer 2 Report

Primary concerns and recommendations

  1. As an RCT with baseline and post-intervention methods, all results should be analyzed as change scores. Ideally, all analyses would also be adjusted for baseline measures.  This can be achieved with repeated measures ANOVA or linear regression. The Methods section lacks an analytic plan suitable for a RCT. The authors generally table the findings nicely by reporting baseline, post-intervention and change scores. However, I believe the manuscript is inconsistent how the outcomes are analyzed, often not presenting change score analysis and not generally adjusting for baseline. The figures also are inconsistent in this regard. This is particularly problematic with the analysis and presentation of CRP and IL-10.

We thank the reviewer for pointing this out, and we have now made an effort to improve our Data Analysis section, clarifying our analytical plan. The main statistical strategy is using percent change (between baseline and 8 weeks) as the basis for our analyses. We calculate this by entering the values for the study end visit (week eight: w8) and the baseline visit (week zero: w0) into the following formula: [(((w8-w0)/w0)*100)]. This creates a baseline correction for our variables. However, we additionally reasoned that comparing the after-treatment effects by “raw values”, for variables where there were no significant differences in baseline scores between the groups, may be interesting and thus included this as well. The analysis is thus consistent with our study protocol approved by the ethical review board. Finally, we decided to highlight any significant findings by figures and leave all the other data displayed in tables, now clearly stated in the Data Analysis section.

Unfortunately, the authors’ response is not adequate.

Ø  The analytic plan now mistakenly applies a 15% change rule to all outcomes.

Ø  There still is no statistical adjustment for the likely covariation between baseline values and pre-post change.

Ø  The analytic plan refers to paired t-tests but once the authors convert the raw data to change scores, there are no paired data.

Ø  All analyses should be done with a since statistical program such as SPSS (not Excel or Prism 7).

Ø  The analysis section does not define statistical significance.

Ø  Please add a statement that no adjustment was made for multiple comparisons.  

Ø  There is no reason to compare the Week 8 values. The proper analysis is being reported as percent change.

Ø  Since CRP is reported in Table 2, please remove it from Table 1.

Ø  Figure 2 is unnecessary. The results are in the tables and are not statistically significant.

  1. Since the study examines the combined effects of a probiotic and omega-3 fatty acids and would like to clarify whether there are additive or synergistic effects, the lack of probiotic only and omega-3 only groups is a serious weakness. This should be explained and discussed in the Introduction, Methods and Discussion.

We realize that we have not clearly stated that the current study is a proof-of-concept study, and we want to express our gratitude to the reviewer for making us aware of this. Proof-of-concept studies are smaller trials that are meant to determine the feasibility of the idea or to verify that the idea will function as envisioned. Here, we were specifically interested in the potential efficacy of combining these supplements. We have now clarified this in the Introduction, Methods, and Discussion/Conclusion of the manuscript and hope this is more to the liking of the reviewer.

I appreciate the authors response and revisions to the manuscript.

  1. The Methods should explain the specific rationale for the chosen probiotic and the dose of fish oil. (It was rather small and, I suspect, smaller than trials upon which this study is based. Why was stool microbiota not an outcome?  The inclusion of vitamin D in the fish oil is problematic since it adds another active ingredient to the intervention arm. Are these weaknesses that warrant more attention in the Discussion?

The omega-3 supplements were provided by GSK/Pfizer; therefore, we were not able to alter the dose.  However, we agree that a higher dose may have elicited more beneficial effects, and we have now noted this in the Discussion.  Stool microbiota was an exploratory outcome, and we are considering to analyze these samples for a future manuscript.  We feel that the inclusion of vitamin D in the fish oil is not too problematic, as the dose is rather low.  However, we agree that this should warrant more attention in planning future studies. 

The authors choice of dose is independent of their choice of producer. Dozens of companies make fish oil available in many doses and ratios. In the Methods, the authors should give a scientific reason for the dose and EPA:DHA ratio they chose for this trial.  

  1. The measure of intestinal permeability is new to me. From the information provided, I cannot understand why the urine collection was separated by hours 0-5 and 6-24. In addition, I do not understand how the data were analyzed. Sucrose is listed as a non-metabolizable sugar, but I believe that is not true. Also, the presented data on sucrose in Table 3 seems to be in error for the active treatment arm: pre-treatment 52.1, post-treatment 50.75, but % change given as 23.5%(!).

We are happy to clarify this further.The sugar recovery ratios from an early urine fraction (urine collection of 0-5 hours after intake of sugars) and a later fraction (urine collection from 6-24 hours after sugar intake) are used to assess the intestinal permeability of the respective intestinal segments. This is possible because the intestinal segments of the upper gastrointestinal tract (evaluated in the 0-5 hour fraction) have shorter transit time and some sugars are absorbed earlier as compared to the lower segments (measured in the 6-24 hour fraction). We added some information that we hope clarifies this in the Methods section and also complemented with a few more references to support this. We have also added a reference in the method section that we hope will help with understanding the data analysis. Indeed, we recognize that the question about non-digestible sugars may be debated, and we hence removed the word non-digestible and hope that this resonates better with our potential readers. Considering Table 3, the displayed data looks strange because we have chosen to show only median values and not arithmetic means (we are doing so because the data is non-normally distributed and non-parametric tests have been used to analyze them). The median looks strange because there is an even number of participants in the treated group and, hence, the median becomes a mean of two numbers (the two most mid ones) which makes the numbers in the table look strange. But we have looked into this before and even re-analyzed this before we realized why the data looks as it does. We hope the reviewer will be satisfied with this response.

Ø  I continue to see erroneous data in Table 3. In the Active Treatment arm, the pre-treatment value is 52.1 and the post-treatment value is 50.75, yet the % change is reported (and later graphed) as +23.6%. Something is clearly wrong.

Ø  Figure 3 lacks a legend or explanation. I see no justification for selectively graphing valeric acid. Figure 3 redundant with Table 3 and can be removed.

  1. The flow chart is good, but the 14 drop-outs require more explanation. Apparently, all enrolled participants gave informed consent and underwent screening procedures to determine eligibility prior to randomization. Therefore, why then did these 14 subjects “decline participation” after randomization? If they dropped out at different steps of the trial, please indicate at roughly what point and for what reason each subject dropped out (e.g., side effects, moved from the area, compliance with study procedures, etc).

Our study was conducted in two cohorts- autumn 2019 and early spring 2020. The majority of our drop-outs were participants that were included and randomized but then decided to drop-out in the few days/weeks before study start as the Covid cases began to increase (Feburary 2020).  Two participants decided to drop-out because they decided they didn’t want to be included, two had issues with other medications/supplements, and one discontinued participation due to side-effects.  We have now included more information about the drop-outs in the beginning of the Results section. 

This information is helpful. However, the text should be revised to not say “declined participation” (since those individuals had agreed to the study and signed consent and were randomized) but that they withdrew from the trial after randomization. Then list the reasons for their withdrawal. The flow chart should be in two columns following randomization so readers can see how many were in each group and the drop-outs in each arm of the trial.

  1. The baseline CRP data in Table 1 indicate a maximum of 18.2 in the intervention group compared to just 3.38 in the control group. I strongly suspect this is due to at least one high outlier in the intervention group. I also suspect that distribution of CRP is highly skewed. How were these distribution issues handled? Such a high baseline CRP strongly suggests an acute inflammatory state which then clearly confounds the interpretation of treatment effects. Perhaps the authors can consider excluding subjects with a high baseline CRP (as is often done in these studies).

We highly appreciate the reviewer pointing out this error.  We had failed to adjust Table 1 for hs-CRP values that excluded this high value that you identified.  We have now used the correct adjusted data set, and the values now properly match what was shown for the baseline data in Table 2. 

It is reasonable to drop the single participant with a very high baseline CRP. However, it also violates the crucial clinical trials precept of “intention to treat.”  If the authors had anticipated this issue, high CRP could have been an exclusion criterion. I might suggest that here the authors describe the issue in the Results and report the primary analysis with and without the high outlier. In addition, CRP values are generally distributed non-normally. Was that the case in this study?  On lines 338-9 the authors refer to sample size calculations being done with a SD estimate of 0.25 for log-level CRP. Yet, I see no indication that CRP was log-transformed prior to analysis. Please address.

  1. The study had many, many outcomes and, therefore, has a high risk of type 1 statistical error. The usual approach is the adjust all secondary analyses to for multiple comparisons. The alternative is repeatedly caution readers that these findings are exploratory and could well be due to chance.

As the study was an exploratory, proof-of-concept study, we have included a wide range of outcomes.  We have taken the reviewer’s suggestion and indicated in several places in the Introduction, Discussion, and Conclusion that the study was exploratory and the findings need to be confirmed in future studies. 

I appreciate the authors response and revisions to the manuscript.

Secondary comments and recommendations

  1. The introduction could be improved by not simply summarizing that probiotics and omega-3 fatty acids seem to reduce inflammation but describe whether and how you posit that their metabolic and physiologic effects would interact. Otherwise, there is not really a scientific rationale to test their “interaction.”

As this is not a study of mechanisms per se but instead a proof-of-concept study to investigate whether this area would be suited for future in-depth studies, we have tried to keep our Introduction at a “depth” of information that we feel reasonable. In the Introduction, we make a case for both supplements to influence immune regulation as well as gut barrier function; from this evidence, we believe that they could act synergistically. We tried to clarify this further by adding the following sentences under Dual Supplementation section in the Introduction: “As outlined above, both probiotics and omega-3 fatty acids have been shown to have similar effects on immune regulation and intestinal barrier function. Yet, very few studies have thus far examined the effects of dual supplementation.”

I appreciate the authors response and revisions to the manuscript.

  1. The authors do a good job of reviewing and citing relevant literatures. However, there is a recent systematic review of the effects of omega-3 fatty acids on sarcopenia they may wish to include -- The effect of long chain omega-3 polyunsaturated fatty acids on muscle mass and function in sarcopenia: A scoping systematic review and meta-analysis. Bird JK, Troesch B, Warnke I, Calder PC.  Clinical Nutrition ESPEN. 46:73-86, 2021 12.

We appreciate the suggestion of this review and find it to be highly interesting; however, in light of suggestions from Reviewer 1, we have removed from the Discussion the paragraph about omega-3/probiotics and physical function.

I appreciate the authors response and revisions to the manuscript.

  1. The Methods includes more detail than is typical. Some could be omitted by either referencing prior reports or placement an online supplement.

The Methods section is now shorter; details for the blood sample collection and analysis and short-chain fatty acids in stool samples have now been moved to supplementary material.  The protocol for the sample preparation and determination of sugar concentration is shorter and now references a previous manuscript. 

I appreciate the authors response and revisions to the manuscript.

  1. The rationale for measuring short-chain fatty acids in stool is not clear to me.

As we briefly outline in the Discussion (bottom of page 18 and top of page 19), SCFAs exert multiple effects on human health such as the inhibition of colonic carcinogenesis, inflammation, and oxidative stress, as well as improvement of colonic barrier function. Hence, SCFAs are metabolites that play significant roles in both immune regulation and intestinal barrier function and are arguably of interest for the current study. 

I appreciate the authors response and revisions to the manuscript.

  1. The fasting insulin and glucose values are typically used to calculate HOMA-IR, which is more informative with respect to insulin resistance than either insulin or glucose alone. Perhaps the authors could report treatment effects on HOMA-IR.

We decided to only include the raw values for glucose and insulin. As there were no significant differences between groups, we decided not to include HOMA-IR values as well. 

The use of fasting insulin and glucose to estimate insulin resistance is done with the HOMA-IR calculation. It should be preformed in this study.

  1. Compliancyshould be changed to compliance.

Each “compliancy” has been changed to “compliance.”

Thanks.

  1. The Discussion considers the putative effects of probiotics on the trial outcomes in great detail without much of any consideration about the putative effects of omega-3 fatty acids.

We have now expanded the Discussion in several places to further consider the putative effects of omega-3 fatty acids on the trial outcomes.   

Thanks.

  1. I do not believe the authors can make any statements about non-significant effects on any outcomes (i.e., trends in the hypothesized direction).  Doing so in Discussion sections generally reflect author bias. In addition, I suspect some of these favorable trends may not persist after consideration of the statistical issues noted above. So, I would ask the authors to generally remove such statements from the Discussion.

We agree with the reviewer and have worked to minimize discussion about favourable trends.  

Thanks.

  1. The Discussion, like the Methods, is too long. Would reduce by 30%.

We agree that the Discussion is too long and have removed two entire paragraphs as well as several sentences in other places. 

Thanks.

  1. There are typographical errors in several places. Careful proofreading by the authors is needed.

We thank the reviewer for noting this. More thorough proofreading has been done, and several errors are now corrected in the text. 

Thanks.

Author Response

  1. As an RCT with baseline and post-intervention methods, all results should be analyzed as change scores. Ideally, all analyses would also be adjusted for baseline measures.  This can be achieved with repeated measures ANOVA or linear regression. The Methods section lacks an analytic plan suitable for a RCT. The authors generally table the findings nicely by reporting baseline, post-intervention and change scores. However, I believe the manuscript is inconsistent how the outcomes are analyzed, often not presenting change score analysis and not generally adjusting for baseline. The figures also are inconsistent in this regard. This is particularly problematic with the analysis and presentation of CRP and IL-10.

We thank the reviewer for pointing this out, and we have now made an effort to improve our Data Analysis section, clarifying our analytical plan. The main statistical strategy is using percent change (between baseline and 8 weeks) as the basis for our analyses. We calculate this by entering the values for the study end visit (week eight: w8) and the baseline visit (week zero: w0) into the following formula: [(((w8-w0)/w0)*100)]. This creates a baseline correction for our variables. However, we additionally reasoned that comparing the after-treatment effects by “raw values”, for variables where there were no significant differences in baseline scores between the groups, may be interesting and thus included this as well. The analysis is thus consistent with our study protocol approved by the ethical review board. Finally, we decided to highlight any significant findings by figures and leave all the other data displayed in tables, now clearly stated in the Data Analysis section.

Unfortunately, the authors’ response is not adequate.

Ø  The analytic plan now mistakenly applies a 15% change rule to all outcomes. We thank the reviewer for identifying this source of confusion.  As we were looking for a percent change in hs-CRP, we wanted to state that we would also look for a percent change in all other variables as well.  We have now removed the phrase “15 percent change” to avoid any confusion that were looking for a 15 percent change for any variable other than hs-CRP.

Ø  There still is no statistical adjustment for the likely covariation between baseline values and pre-post change. We thank the reviewer for this suggestion. However, as we have taken into account this issues by comparing values at baseline, we do not feel that it is necessary to perform this type of analysis. 

Ø  The analytic plan refers to paired t-tests but once the authors convert the raw data to change scores, there are no paired data. Paired t-tests were included in our original analysis plan in the event that we made these comparisions.  As we have not included that analysis in this manuscript, we agree that paired t-tests should not be mentioned in the methods section.  We thank the reviewer for this suggestion.

Ø  All analyses should be done with a since statistical program such as SPSS (not Excel or Prism 7). We do agree that Excel should not be mentioned here. Excel was in some instances used to digitalize and organize data prior to analysis, but we have now removed Excel from the text. GraphPad Prism is, however, commonly used and accepted for scientific data analysis by Swedish universities, and we hope that the reviewer is prepared to accept this.

Ø  The analysis section does not define statistical significance. We thank the reviewer for pointing this out and have now added this information at the end of the data analysis section.

Ø  Please add a statement that no adjustment was made for multiple comparisons.  We agree with the reviewer and have added this statement to the data analysis section. 

Ø  There is no reason to compare the Week 8 values. The proper analysis is being reported as percent change. We do agree with the reviewer on this point, considering RCT studies. However, as this study is a smaller proof-of-concept study, we allowed ourselves to do these statistical comparisons for potential guidance in future studies. We wished to exhaust the opportunities to detect even subtle changes, reasoning that if a number of small indications point in a similar direction it may be worthwhile to pursue this area of investigation in future trials. We hope that the reviewer can grant us this freedom. Revising the manuscript by removing these comparisons would at this point require much more time for substantial revisions. In recognition of this critique, we have, however, revised the manuscript at a number of places to make sure that we do not make “too much” of these comparisons.

Ø  Since CRP is reported in Table 2, please remove it from Table 1.  We have removed this value from Table 1. 

Ø  Figure 2 is unnecessary. The results are in the tables and are not statistically significant.  We have removed Figure 2 and 3 from the manuscript as the reviewer suggested.

  1. Since the study examines the combined effects of a probiotic and omega-3 fatty acids and would like to clarify whether there are additive or synergistic effects, the lack of probiotic only and omega-3 only groups is a serious weakness. This should be explained and discussed in the Introduction, Methods and Discussion.

We realize that we have not clearly stated that the current study is a proof-of-concept study, and we want to express our gratitude to the reviewer for making us aware of this. Proof-of-concept studies are smaller trials that are meant to determine the feasibility of the idea or to verify that the idea will function as envisioned. Here, we were specifically interested in the potential efficacy of combining these supplements. We have now clarified this in the Introduction, Methods, and Discussion/Conclusion of the manuscript and hope this is more to the liking of the reviewer.

I appreciate the authors response and revisions to the manuscript.

  1. The Methods should explain the specific rationale for the chosen probiotic and the dose of fish oil. (It was rather small and, I suspect, smaller than trials upon which this study is based. Why was stool microbiota not an outcome?  The inclusion of vitamin D in the fish oil is problematic since it adds another active ingredient to the intervention arm. Are these weaknesses that warrant more attention in the Discussion?

The omega-3 supplements were provided by GSK/Pfizer; therefore, we were not able to alter the dose.  However, we agree that a higher dose may have elicited more beneficial effects, and we have now noted this in the Discussion.  Stool microbiota was an exploratory outcome, and we are considering to analyze these samples for a future manuscript.  We feel that the inclusion of vitamin D in the fish oil is not too problematic, as the dose is rather low.  However, we agree that this should warrant more attention in planning future studies. 

The authors choice of dose is independent of their choice of producer. Dozens of companies make fish oil available in many doses and ratios. In the Methods, the authors should give a scientific reason for the dose and EPA:DHA ratio they chose for this trial.  

We agree that there are many fish oil and probiotic supplements on the market and that one could reason that the lack of effect in this study is due to too low doses, which we mention briefly in our discussion: “The lack of significant hs-CRP decreases in the current study may suggest that not all probiotic strains are efficacious in reducing low-grade inflammation in the elderly. Furthermore, a higher dose of omega-3 may have also been necessary to see positive effects.” There are, however, scientific studies showing effects also from doses similar to the one used in the current study, e.g. Watson H, et al. A randomised trial of the effect of omega-3 polyunsaturated fatty acid supplements on the human intestinal microbiota. Gut. 2018 Nov;67(11):1974-1983. Furthermore, there is still poor agreement between recommendation of daily nutritional intake and doses used in interventional research studies. Here, we aimed to perform a study testing a supplement and a dose that would be readily available to the consumers if the intervention showed promising results. Also, we reasoned similiarly in selecting the probiotic strains, valuing the fact that the product is readily available and composed of well-known and characterized bacterial strains, in doses that are commonly recommended. Indeed, we systematically reviewed this topic last year (Hutchinson AN, Bergh C, Kruger K, Sűsserová M, Allen J, Améen S, Tingö L. The Effect of Probiotics on Health Outcomes in the Elderly: A Systematic Review of Randomized, Placebo-Controlled Studies. Microorganisms. 2021 Jun 21;9(6):1344), concluding that there is still insufficient evidence to determine if a particular probiotic/synbiotic combination or duration of treatment is efficacious in improving health outcomes in healthy elderly. We hope that the reviewer can accept that we arrived at these doses and strains.

  1. The measure of intestinal permeability is new to me. From the information provided, I cannot understand why the urine collection was separated by hours 0-5 and 6-24. In addition, I do not understand how the data were analyzed. Sucrose is listed as a non-metabolizable sugar, but I believe that is not true. Also, the presented data on sucrose in Table 3 seems to be in error for the active treatment arm: pre-treatment 52.1, post-treatment 50.75, but % change given as 23.5%(!).

We are happy to clarify this further.The sugar recovery ratios from an early urine fraction (urine collection of 0-5 hours after intake of sugars) and a later fraction (urine collection from 6-24 hours after sugar intake) are used to assess the intestinal permeability of the respective intestinal segments. This is possible because the intestinal segments of the upper gastrointestinal tract (evaluated in the 0-5 hour fraction) have shorter transit time and some sugars are absorbed earlier as compared to the lower segments (measured in the 6-24 hour fraction). We added some information that we hope clarifies this in the Methods section and also complemented with a few more references to support this. We have also added a reference in the method section that we hope will help with understanding the data analysis. Indeed, we recognize that the question about non-digestible sugars may be debated, and we hence removed the word non-digestible and hope that this resonates better with our potential readers. Considering Table 3, the displayed data looks strange because we have chosen to show only median values and not arithmetic means (we are doing so because the data is non-normally distributed and non-parametric tests have been used to analyze them). The median looks strange because there is an even number of participants in the treated group and, hence, the median becomes a mean of two numbers (the two most mid ones) which makes the numbers in the table look strange. But we have looked into this before and even re-analyzed this before we realized why the data looks as it does. We hope the reviewer will be satisfied with this response.

Ø  I continue to see erroneous data in Table 3. In the Active Treatment arm, the pre-treatment value is 52.1 and the post-treatment value is 50.75, yet the % change is reported (and later graphed) as +23.6%. Something is clearly wrong.  Thank you for persistently pointing this out. We now found the error and have corrected the data accordingly. As there are no differences between the two groups, we updated the results and discussion accordingly.

Ø  Figure 3 lacks a legend or explanation. I see no justification for selectively graphing valeric acid. Figure 3 redundant with Table 3 and can be removed. On the basis of this comment, we decided to remove Figure 3.

  1. The flow chart is good, but the 14 drop-outs require more explanation. Apparently, all enrolled participants gave informed consent and underwent screening procedures to determine eligibility prior to randomization. Therefore, why then did these 14 subjects “decline participation” after randomization? If they dropped out at different steps of the trial, please indicate at roughly what point and for what reason each subject dropped out (e.g., side effects, moved from the area, compliance with study procedures, etc).

Our study was conducted in two cohorts- autumn 2019 and early spring 2020. The majority of our drop-outs were participants that were included and randomized but then decided to drop-out in the few days/weeks before study start as the Covid cases began to increase (Feburary 2020).  Two participants decided to drop-out because they decided they didn’t want to be included, two had issues with other medications/supplements, and one discontinued participation due to side-effects.  We have now included more information about the drop-outs in the beginning of the Results section. 

This information is helpful. However, the text should be revised to not say “declined participation” (since those individuals had agreed to the study and signed consent and were randomized) but that they withdrew from the trial after randomization. Then list the reasons for their withdrawal. The flow chart should be in two columns following randomization so readers can see how many were in each group and the drop-outs in each arm of the trial.  We changed the text to declined/discontinued participation as the reviewer suggested.  We have now listed the reasons in the results text.  We also now included in Figure 1 the number in each arm that discontinued participation. 

  1. The baseline CRP data in Table 1 indicate a maximum of 18.2 in the intervention group compared to just 3.38 in the control group. I strongly suspect this is due to at least one high outlier in the intervention group. I also suspect that distribution of CRP is highly skewed. How were these distribution issues handled? Such a high baseline CRP strongly suggests an acute inflammatory state which then clearly confounds the interpretation of treatment effects. Perhaps the authors can consider excluding subjects with a high baseline CRP (as is often done in these studies).

We highly appreciate the reviewer pointing out this error.  We had failed to adjust Table 1 for hs-CRP values that excluded this high value that you identified.  We have now used the correct adjusted data set, and the values now properly match what was shown for the baseline data in Table 2. 

It is reasonable to drop the single participant with a very high baseline CRP. However, it also violates the crucial clinical trials precept of “intention to treat.”  If the authors had anticipated this issue, high CRP could have been an exclusion criterion. I might suggest that here the authors describe the issue in the Results and report the primary analysis with and without the high outlier. In addition, CRP values are generally distributed non-normally. Was that the case in this study?  On lines 338-9 the authors refer to sample size calculations being done with a SD estimate of 0.25 for log-level CRP. Yet, I see no indication that CRP was log-transformed prior to analysis. Please address. We thank the reviewer for the continued discussion over this issue.  In order to display the data consistently, we have only now included the intention to treat set.  Including or excluding the outlier does not change any of the conclusions of the data, as the percent change values are insignificant either way.  We hope this satisfies the reviewer.  Yes, indeed, our CRP values did not follow a normal distribution, and we used the appropriate statistical analysis for this.  In our lab and other groups, we utilize a formal for power calculations with log-level CRP.  However, our primary outcome was a 15% change in hs-CRP, so we decided to analyze the data with % changes instead of log-transformation. 

  1. The study had many, many outcomes and, therefore, has a high risk of type 1 statistical error. The usual approach is the adjust all secondary analyses to for multiple comparisons. The alternative is repeatedly caution readers that these findings are exploratory and could well be due to chance.

As the study was an exploratory, proof-of-concept study, we have included a wide range of outcomes.  We have taken the reviewer’s suggestion and indicated in several places in the Introduction, Discussion, and Conclusion that the study was exploratory and the findings need to be confirmed in future studies. 

I appreciate the authors response and revisions to the manuscript.

Secondary comments and recommendations

  1. The introduction could be improved by not simply summarizing that probiotics and omega-3 fatty acids seem to reduce inflammation but describe whether and how you posit that their metabolic and physiologic effects would interact. Otherwise, there is not really a scientific rationale to test their “interaction.”

As this is not a study of mechanisms per se but instead a proof-of-concept study to investigate whether this area would be suited for future in-depth studies, we have tried to keep our Introduction at a “depth” of information that we feel reasonable. In the Introduction, we make a case for both supplements to influence immune regulation as well as gut barrier function; from this evidence, we believe that they could act synergistically. We tried to clarify this further by adding the following sentences under Dual Supplementation section in the Introduction: “As outlined above, both probiotics and omega-3 fatty acids have been shown to have similar effects on immune regulation and intestinal barrier function. Yet, very few studies have thus far examined the effects of dual supplementation.”

I appreciate the authors response and revisions to the manuscript.

  1. The authors do a good job of reviewing and citing relevant literatures. However, there is a recent systematic review of the effects of omega-3 fatty acids on sarcopenia they may wish to include -- The effect of long chain omega-3 polyunsaturated fatty acids on muscle mass and function in sarcopenia: A scoping systematic review and meta-analysis. Bird JK, Troesch B, Warnke I, Calder PC.  Clinical Nutrition ESPEN. 46:73-86, 2021 12.

We appreciate the suggestion of this review and find it to be highly interesting; however, in light of suggestions from Reviewer 1, we have removed from the Discussion the paragraph about omega-3/probiotics and physical function.

I appreciate the authors response and revisions to the manuscript.

  1. The Methods includes more detail than is typical. Some could be omitted by either referencing prior reports or placement an online supplement.

The Methods section is now shorter; details for the blood sample collection and analysis and short-chain fatty acids in stool samples have now been moved to supplementary material.  The protocol for the sample preparation and determination of sugar concentration is shorter and now references a previous manuscript. 

I appreciate the authors response and revisions to the manuscript.

  1. The rationale for measuring short-chain fatty acids in stool is not clear to me.

As we briefly outline in the Discussion (bottom of page 18 and top of page 19), SCFAs exert multiple effects on human health such as the inhibition of colonic carcinogenesis, inflammation, and oxidative stress, as well as improvement of colonic barrier function. Hence, SCFAs are metabolites that play significant roles in both immune regulation and intestinal barrier function and are arguably of interest for the current study. 

I appreciate the authors response and revisions to the manuscript.

  1. The fasting insulin and glucose values are typically used to calculate HOMA-IR, which is more informative with respect to insulin resistance than either insulin or glucose alone. Perhaps the authors could report treatment effects on HOMA-IR.

We decided to only include the raw values for glucose and insulin. As there were no significant differences between groups, we decided not to include HOMA-IR values as well. 

The use of fasting insulin and glucose to estimate insulin resistance is done with the HOMA-IR calculation. It should be preformed in this study.  We have now updated our manuscript to include HOMA-IR. 

  1. Compliancyshould be changed to compliance.

Each “compliancy” has been changed to “compliance.”

Thanks.

  1. The Discussion considers the putative effects of probiotics on the trial outcomes in great detail without much of any consideration about the putative effects of omega-3 fatty acids.

We have now expanded the Discussion in several places to further consider the putative effects of omega-3 fatty acids on the trial outcomes.   

Thanks.

  1. I do not believe the authors can make any statements about non-significant effects on any outcomes (i.e., trends in the hypothesized direction).  Doing so in Discussion sections generally reflect author bias. In addition, I suspect some of these favorable trends may not persist after consideration of the statistical issues noted above. So, I would ask the authors to generally remove such statements from the Discussion.

We agree with the reviewer and have worked to minimize discussion about favourable trends.  

Thanks.

  1. The Discussion, like the Methods, is too long. Would reduce by 30%.

We agree that the Discussion is too long and have removed two entire paragraphs as well as several sentences in other places. 

Thanks.

  1. There are typographical errors in several places. Careful proofreading by the authors is needed.

We thank the reviewer for noting this. More thorough proofreading has been done, and several errors are now corrected in the text. 

Thanks.

Round 3

Reviewer 2 Report

The authors have satisfied my concerns with one minor exception. Whereas the authors state they added a statement about the lack of adjustment for multiple comparisons, I do no see it in the Data Analysis section.

Ø  Please add a statement that no adjustment was made for multiple comparisons.  We agree with the reviewer and have added this statement to the data analysis section. 

Author Response

We are very pleased to hear that we have satisfied your concerns.  We have now added a sentence to the Data analysis section that reads: 

"As the study was an initial investigation, no adjustments were made for multiple comparisons."

Thank you so much!